# General self-efficacy in individuals with visual impairment compared with the general population

**Audun Brunes** [1]*, **Marianne B. Hansen** [2], **Trond Heir** [1,3]

1 Section for Trauma, Catastrophes and Forced Migration—Adults and Elderly, Norwegian Centre for Violence and Traumatic Stress Studies, Oslo, Norway, 2 Division of Mental Health and Addiction, Norwegian National Unit for Hearing Impairment and Mental Health, Oslo University Hospital, Oslo, Norway, 3 Institute of Clinical Medicine, Faculty of Medicine, University of Oslo, Oslo, Norway

☉ These authors contributed equally to this work.

* audun.brunes@nkvts.no

## Abstract

### Background

Knowledge about self-efficacy and its significance for the quality of life of people with visual impairment is lacking. The aims of the study were to compare general self-efficacy in individuals with visual impairment with the general population, and to investigate the association between self-efficacy and life satisfaction.

### Methods

A telephone-based cross-sectional survey was conducted between January and May 2017 in a probability sample of adults who were members of the Norwegian Association of the Blind and Partially Sighted. Participants were asked questions about their sociodemographic characteristics, characteristics of vision loss, general self-efficacy (General Self-efficacy Scale), and life satisfaction (Cantril's Ladder of Life Satisfaction). We obtained norm data from a representative survey of the general Norwegian population (N = 1792; mean age 53.2 years; 52.5% females).

### Results

People with visual impairment had higher levels of general self-efficacy than people in the general population (Mean: 31.5 versus 29.0, p < .001). Results from linear regression analyses of the visual impairment population showed that higher education and residential in an urban municipality were associated with higher self-efficacy. Having additional impairments and a previous history of physical or sexual assaults were associated with lower self-efficacy. A linear dose-response relationship was found between self-efficacy and life satisfaction, in the visual impairment population as well as in the general population.

**Data Availability Statement:** Data are from the research project European Network for Psychosocial Crisis Management – Assisting Disabled in Case of Disaster (EUNAD). According to the approval from the Norwegian Regional

Ethical Committee, data is to be stored properly and in line with the Norwegian privacy protection laws. Data contains sensitive information from a small group. Public availability may result in the possibility of indirect identification, and thus would compromise privacy of the participants. However, the data is freely available to interested researchers upon request, pending ethical approval from our ethical committee: post@helseforskning.etikk.no.

**Funding:** This work was supported by grants from the European Commission, Directorate-General for European Civil Protection and Humanitarian Aid Operations (TH: No. ECHO/SUB/2015/718665/ PREP17) and three grants from the Norwegian Association of the Blind and Partially Sighted (AB: No. S23/2017; AB: No. S20/2018; AB: No. S12/ 2019). The funders had no role in study design, data collection and analysis, decision to publish, or preparation of the manuscript.

**Competing interests:** I have read the journal's policy and the authors of this manuscript have the following competing interests: Research grants from the European Commission and research grants from the Norwegian Association of the Blind and Partially Sighted. This does not alter our adherence to PLOS ONE policies on sharing data and materials.

## Conclusions

People with visual impairment have higher self-efficacy than people in the general population, possibly due to extensive mastery experience in how to handle life as visually impaired. Self-efficacy seems to be important in achieving the best possible life.

## Introduction

Visual impairment is a common condition, affecting about one billion of the world's population, and has a diverse set of causes, severities, and progression rates [1]. In general, visual impairment represents a limitation in access to information, in interaction with the environment, and in orientation and mobility. People with visual impairment are more prone to loneliness [2] and depression [3]. Those who have greater visual impairments report less control over their life and environment, and they have lower quality of life [4, 5]. It has been suggested that visual impairment may increase psychological distress or reduce quality of life by diminishing psychological resources such as self-efficacy [6].

Self-efficacy refers to a person's belief in his or her ability to successfully perform a task or influence an event to a desired outcome [7, 8]. The self-efficacy concept is usually tied less to specific tasks or situations, but reflects a person's ability to feel confident and rely on his or her own efforts to cope with the challenges of life in general [8]. According to Bandura [7], a person with high self-efficacy considers a problem more as a challenge that can be overcome, rather than an obstacle or threat. From a health perspective, higher self-efficacy is associated with the ability to improve health through rehabilitation from illness [9–11], smoking cessation [12], or adherence to recommended physical activity [13] or diet [14]. Also for people with visual impairment, there are indications that self-efficacy may be related to better adjustment, health outcomes, and quality of life [15].

Self-efficacy may be developed through successful experience, social interactions, or individual emotional or physical reactions, of which successful experience is the most effective procedure [7]. In line with this theory, we might expect general self-efficacy to develop with increasing experience in mastering the extensive challenges usually faced by people with visual impairment. On the other hand, general attitudes towards people with low vision or blindness, as well as limited actual opportunities to succeed to the same degree as people without vision loss, may have limited their belief in mastery.

Few studies have compared self-efficacy among the visually impaired with the general population. In studies of college students, self-efficacy was largely unrelated to visual impairment [16] or other disabilities [17]. However, we have been unable to identify studies aimed at examining general self-efficacy in people with and without visual impairment in more comprehensive population studies.

Individuals with higher general self-efficacy are more satisfied with their life [18]. The relationship may be modified through success in various aspects of life, such as education, work, family, or social adjustment. People with visual impairment are more prone to several conditions that have been shown to be associated with lower life satisfaction such as bullying [19], abuse [20], loneliness [2], or mental disorders [3, 15]. Therefore, it is uncertain whether the relationship between self-efficacy and quality of life is the same in people with visual impairment as in the general population.

The aim of this cross-sectional study was to examine general self-efficacy in people with visual impairment compared with the general population. A second aim was to examine the

association between self-efficacy and life satisfaction in people with visual impairment versus the general population.

## Materials and methods

### Design and participants

**Visual impairment population.**   This cross-sectional telephone survey included a probability sample of adult members of the Norwegian Association of the Blind and Partially Sighted. All members aged 18 years or older were eligible to participate if they had a diagnosis of visual impairment or progressive eye condition, and were able to speak and understand the Norwegian language. Most members are of middle-age or older. We therefore used age-stratified sampling to allow for more precise estimations across the different age groups. First, the study population was divided into four age groups (years: 18–35, 36–50, 51–65, ≥ 66), and then we attempted to survey a random sample of about 200 participants from each age group. Of the 1216 individuals who were contacted, 736 (61%) participated by completing the interview. A flow chart of the sample selection is provided elsewhere [19].

**General population.**   We extracted norm data from the Norwegian Population Study (Nor-Pop), a cross-sectional survey including a representative sample of non-institutionalized Norwegian adults aged 18 years or older [21]. Simple random sampling was conducted based on names and addresses from the Central National Register of Norway, and efforts were made to ensure that the sample reflected the Norwegian population in terms of age, gender, and geographical location. Data were collected between 2014 and 2015 by postal questionnaires. Of the 5500 people selected for participation, nine persons had died, 21 were not able to fill out the questionnaire, and 499 envelopes had non-valid addresses. This resulted in a total of 4971 eligible individuals, of which 1792 (36%) participated by completing and returning the questionnaire.

### Measurements

**General self-efficacy.**   The General Self-efficacy Scale measures optimistic self-belief in coping with the demands, tasks and challenges of life in general [22]. It consists of 10 statements that respondents rate on a scale from 1 (not at all true) to 4 (exactly true). Examples of statements are "I can always manage to solve difficult problems if I try hard enough" (item 1) and "I am certain that I can accomplish my goals" (item 3). The responses on each statement are then summed up to a total score, ranging from 10–40, with higher scores indicating higher general self-efficacy. High correlations with measures of self-appraisal, self-acceptance, and optimism, and with social-cognitive variables, behavior-specific self-efficacy, and well-being [23], have indicated theoretical accuracy of the concept. The scale has been translated into Norwegian and found to have acceptable construct/content validity [21]. For the main analyses, we treated general self-efficacy as an untransformed continuous variable.

**Life satisfaction.**   In the visual impairment population, the Cantril's Ladder of Life Satisfaction was used to measure the participant's general satisfaction with life [24]. The participants were asked to imagine a ladder with 10 steps, with the bottom step representing the worst possible life (a score of 1) and the top step representing the best possible life (a score of 10). In the general population, the participants' life satisfaction was assessed by one item retrieved from the European Organization for Research and Treatment of Cancer (EORTC) [25]. The participants were asked to respond to the question "How has your quality of life been during the last week?" The response format was a scale from 0 (extremely poor) to 10 (excellent). Both life satisfaction variables were treated as untransformed continuous variables in the main analyses.

**Independent variables.**   In both the visual impairment population and the general population, assessments were made regarding sociodemographic characteristics, including age (years:

18–35, 36–50, 51–65, $\geq$ 66), gender, education (years: $< 14$, $\geq 14$), marital status (married/cohabitant, unmarried), occupational status (employed, unemployed, retired), and place of residence (rural areas, urban areas).

In the visual impairment population, we also included information about past experiences with bullying or physical or sexual assaults (none, bullying only, assaults), having additional impairments (no, yes), and the degree of vision loss (moderate VI/other, severe VI, blindness). Additionally, we created an 'age of VI onset' variable by subtracting the participant's age with the number of years since VI onset. The variable was categorized into the following three categories: 'congenital', 'childhood/adolescence (1–24 years)', and 'adulthood ($\geq$ 25 years)'.

## Statistical analysis

We used Stata Version 15 (Stata Corp., Texas, USA) for all statistical analyses. The significance level was set at p = .05. Descriptive statistics included histograms, means, standard deviations (SDs), frequencies, and percentages. We tabulated each study characteristic separately for the visual impairment population and the general population and used Pearson's Chi-squared statistics to test for differences in frequency counts. Additionally, we used independent sample t-tests to calculate differences in mean self-efficacy scores between the visual impairment population and the general population according to the participants' age (age groups: 18–35, 36–50, 51–65, and $\geq$ 66) and gender.

We applied linear regression analyses to calculate beta-values and 95% confidence intervals (CIs) for the independent associations of general self-efficacy with participants' sociodemographic, VI characteristics, and past exposure to bullying and physical or sexual assaults. Occupational status was left out from the analysis due to its strong correlation with age (r = .68). Our data met all assumptions relating to linear regression, and we did not find any impact from outliers on the main results.

We also used linear regression models to examine the association between general self-efficacy and life satisfaction for both the visual impairment population and the general population. Results from the likelihood ratio test showed a dose-response relationship between self-efficacy and life satisfaction for both the visual impairment population ($\chi^2$: 5.5, p = .23) and the general population ($\chi^2$: 3.3, p = .52). We therefore decided to treat self-efficacy as a continuous variable in the regression model. The models were either unadjusted or adjusted for age, gender, education, marital status, and place of residence.

## Ethics

The Regional Committee for Medical and Health Research Ethics was sought and they confirmed that the study required no formal ethical approval as it was carried out in accordance with principles of anonymized data (Reference number: 2016/1615A). The participants were informed about all aspects of the project, including potential risks and the voluntary nature of the survey, and consented by completing the survey. No identifying information was collected. Written consent was not obtained to secure anonymity of the participants.

## Results

### Sample characteristics

The statistical analyses included 736 participants with visual impairment and 1792 participants from the general population. In both surveys, non-participants were more likely than participants to be of young or old age [19, 21]. The visual impairment population had no sources of missing data; all participants answered all questions and none chose to withdraw from the

study. For the general population, the percentage of missing data ranged between 0% and 2% across the different variables.

Table 1 shows characteristics of males and females from the visual impairment population and the general population. Male and female participants with visual impairment were more likely to be unemployed and unmarried compared with those from the general population. Additionally, for males, visually impaired people were younger than that observed in the general population. For females, lower levels of education and being residential in a rural municipality were more often reported by people with visual impairment compared with people from the general population (p < .05).

For people with visual impairment, forty-three percent had congenital vision loss and the remaining 57% had acquired vision loss during childhood or adulthood. Roughly one in three (35%) participants had other impairments in addition to the vision loss. The onset-age of vision loss ranged from 0 to 76 years (Mean: 19 years). Thirty-five percent were moderately impaired, 40% were severely impaired, and 25% were blind.

## General self-efficacy

The distribution of the General Self-efficacy Scale in the visual impairment population and in the general population is graphed in Fig 1. The sample mean scores on self-efficacy were higher in the visual impairment population compared to the general population (31.5 versus 29.0, p < .001). The difference between the two populations was present across all age groups, and particularly among the oldest participants (Table 2).

Associated factors of self-efficacy in the visual impairment population are shown in Table 3. The results from the univariable regression models showed that lower education, unmarried, being residential in rural areas, having additional impairments, and a previous history of physical or sexual assaults were associated with lower self-efficacy. All factors, except for marital status, remained statistically significantly associated with self-efficacy in the multivariable

**Table 1. Characteristics of the visual impairment population and the general population, according to participant's gender.**

| Characteristics | VI male (n = 333)[b] n (%) | GP male (n = 834)[b] n (%) | p-value | VI female (n = 403)[b] n (%) | GP female (n = 945)[b] n (%) | p-value |
|---|---|---|---|---|---|---|
| **Age (years):** 18–35 | 69 (20.7) | 105 (12.7) | .001 | 88 (21.8) | 189 (20.1) | .20 |
| 36–50 | 85 (25.5) | 184 (22.2) | | 101 (25.1) | 273 (29.0) | |
| 51–65 | 94 (28.2) | 286 (34.5) | | 106 (26.3) | 267 (28.4) | |
| ≥ 66 | 85 (25.5) | 253 (30.6) | | 108 (26.8) | 212 (22.5) | |
| **Education (years):** < 14 | 170 (51.0) | 398 (48.0) | .34 | 231 (57.3) | 425 (45.1) | < .001 |
| ≥ 14 | 163 (49.0) | 432 (52.0) | | 172 (42.7) | 517 (54.9) | |
| **Occupational status:** Employed[c] | 152 (45.7) | 526 (63.4) | < .001 | 143 (35.5) | 641 (68.3) | < .001 |
| Unemployed | 108 (32.4) | 60 (7.2) | | 163 (40.5) | 82 (8.7) | |
| Retired | 73 (21.9) | 244 (29.4) | | 97 (24.1) | 216 (23.0) | |
| **Marital status:** Married/cohabitant | 166 (49.9) | 672 (80.9) | < .001 | 181 (44.9) | 706 (75.2) | < .001 |
| Unmarried[d] | 167 (50.1) | 159 (19.1) | | 222 (55.1) | 233 (24.8) | |
| **Place of residence:** Rural areas | 172 (51.6) | 400 (48.4) | .32 | 227 (56.3) | 444 (47.3) | .003 |
| Urban areas | 161 (48.4) | 426 (51.6) | | 176 (43.7) | 494 (52.7) | |

[a]Abbreviations: VI visual impairment; GP general population.

[b]The VI population had no missing data. The number of women or men from the general population with missing data ranged between three and nine.

[c]The employed category encompassed people reporting to be in work, under education, or in military service.

[d]Unmarried involved those who were single, divorced, or widowed.

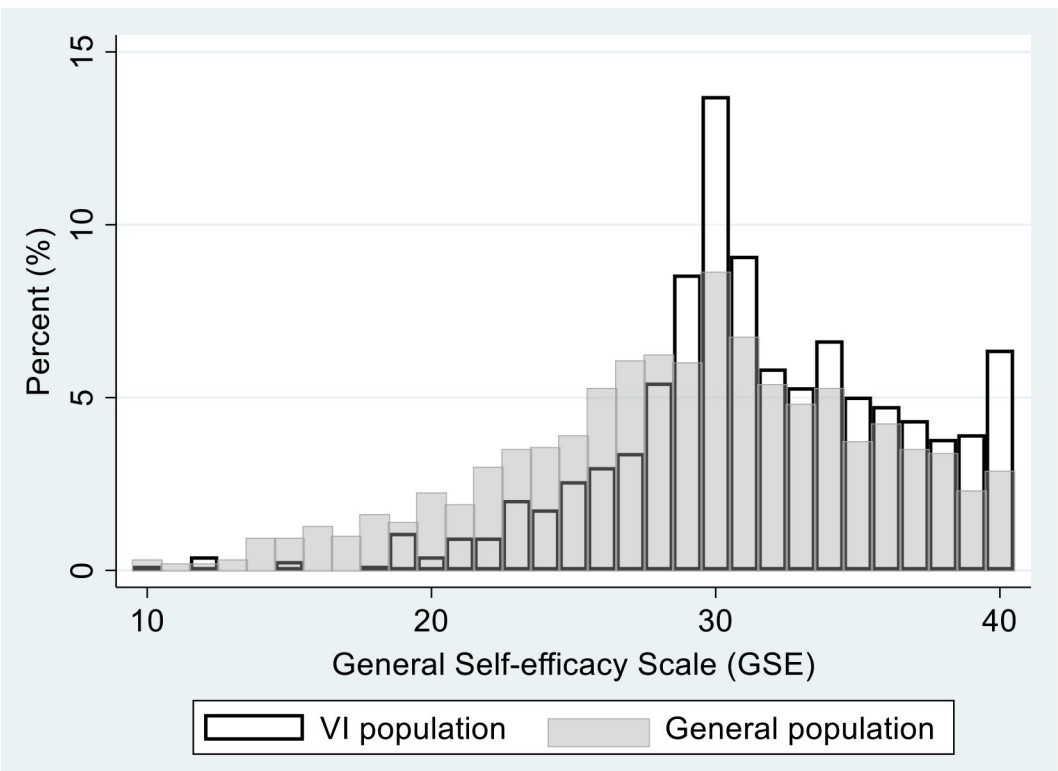

**Fig 1. General self-efficacy in people with visual impairment (n = 736) and in the general population (n = 1792).**

model. Self-efficacy was not related to age, gender, degree of impairment, or onset-age of vision loss.

**Table 2. Independent t-test comparing mean general self-efficacy score between the visual impairment population and the general population, according to participant's age and gender.**

| | | General Self-efficacy Scale[b] | | | |
|---|---|---|---|---|---|
| | | VI (n = 736) | GP (n = 1792) | | |
| | N | Mean (SD) | Mean (SD) | Difference (95% CI) | p-value |
| **Male** | | | | | |
| **Age (years): 18–35** | 173 | 31.8 (5.2) | 31.8 (5.0) | 0.03 (−1.51, 1.58) | .97 |
| 36–50 | 269 | 31.3 (5.3) | 30.2 (5.9) | 1.11 (−0.35, 2.58) | .14 |
| 51–65 | 376 | 32.3 (4.8) | 29.6 (5.7) | 2.67 (1.39, 3.84) | < .001 |
| ≥ 66 | 333 | 31.9 (5.3) | 28.4 (6.2) | 3.50 (2.01, 4.98) | < .001 |
| All age groups | 1157 | 31.8 (5.1) | 29.6 (5.9) | 2.18 (1.46, 2.90) | < .001 |
| **Female** | | | | | |
| **Age (years): 18–35** | 273 | 31.1 (4.9) | 28.7 (6.5) | 2.32 (0.78, 3.85) | .003 |
| 36–50 | 370 | 30.7 (5.6) | 29.1 (6.1) | 1.63 (0.26, 3.00) | .02 |
| 51–65 | 365 | 31.8 (4.9) | 28.4 (6.1) | 3.35 (2.04, 4.66) | < .001 |
| ≥ 66 | 310 | 31.3 (4.9) | 27.3 (6.9) | 4.01 (2.55, 5.48) | < .001 |
| All age groups | 1322 | 31.2 (5.1) | 28.4 (6.4) | 2.79 (2.09, 3.50) | < .001 |

[a]Abbreviations: VI visual impairment; GP general population; CI confidence interval; SD standard deviation.

[b]All participants from the visual impairment population responded to the questions about age, gender and self-efficacy, whereas 59 participants had missing data from the general population.

Table 3. Univariable and multivariable regression analyses of factors associated with general self-efficacy in the visual impairment population (n = 736).

| | Univariable[b] | | Multivariable[b] | |
|---|---|---|---|---|
| | Beta (95% CI) | p-value | Beta (95%CI) | p-value |
| **Age (continuous)** | 0.13 (−0.09, 0.34) | .25 | −0.02 (−0.26, 0.22) | .87 |
| **Gender:** Male | Referent | | Referent | |
| Female | −0.59 (−1.33, 0.15) | .12 | −0.28 (−1.00, 0.44) | .45 |
| **Education (years):** ≥ 14 | Referent | | Referent | |
| < 14 | **−0.98 (−1.72, −0.25)** | **.009** | **−0.77 (−1.50, −0.04)** | **.04** |
| **Marital status:** Married/cohabitant | Referent | | Referent | |
| Unmarried | **−1.17 (−1.90, −0.44)** | **.002** | −0.65 (−1.39, 0.09) | .08 |
| **Place of residence:** Urban areas | Referent | | Referent | |
| Rural areas | **−1.71 (−2.44, −0.99)** | **< .001** | **−1.66 (−2.41, −0.92)** | **< .001** |
| **Other impairments:** No | Referent | | Referent | |
| Yes | **−2.13 (−2.89, −1.38)** | **< .001** | **−1.86 (−2.63, −1.10)** | **< .001** |
| **VI severity**: Moderate/other | Referent | | Referent | |
| Severe | 0.23 (−0.62, 1.08) | .60 | 0.34 (−0.50, 1.18) | .43 |
| Blind | −0.25 (−1.21, 0.72) | .61 | −0.08 (−1.05, 0.89) | .87 |
| **Age at VI onset:** Congenital | Referent | | Referent | |
| Childhood or youth | 0.47 (−0.53, 1.47) | .36 | 0.68 (−0.29, 1.65) | .17 |
| Adulthood | 0.64 (−0.19, 1.46) | .13 | 0.40 (−0.50, 1.31) | .38 |
| **Adverse exposure:** None | Referent | | Referent | |
| Bullying only | −0.51 (−1.37, 0.35) | .25 | −0.18 (−1.02, 0.73) | .75 |
| Assaults | **−1.48 (−2.43, −0.53)** | **.002** | **−0.99 (−1.95, −0.03)** | **.04** |

[a]Abbreviations: VI visual impairment; CI confidence interval; SD standard deviation.

[b]Results in bold indicates statistical significance.

### Associations between self-efficacy and life satisfaction

Fig 2 displays the linear associations between general self-efficacy and life satisfaction for people with visual impairment and the general population. Results from unadjusted regression models showed that higher scores on the General Self-efficacy Scale were associated with higher levels of life satisfaction, both for people with visual impairment (β: 0.14, 95% CI: 0.12, 0.17) and the general population (β: 0.11, 95% CI: 0.09, 0.13). No changes in estimates were found after adjustments for age, gender, education, marital status, and place of residence (visual impairment: β: 0.14; general population: β: 0.11).

## Discussion

Findings from the present study showed that the population of blind and partially sighted had higher levels of general self-efficacy than people in the general population. For people with visual impairment, higher self-efficacy was associated with higher education and residential in more urban areas, whereas lower levels of self-efficacy were observed among those who had additional impairments or past experiences with physical or sexual assaults. We found a linear dose-response relationship between self-efficacy and life satisfaction in both populations.

### General self-efficacy

The higher levels of general self-efficacy in the visual impairment population were intriguing, given the greater barriers that people with visual impairment may face in terms of professional

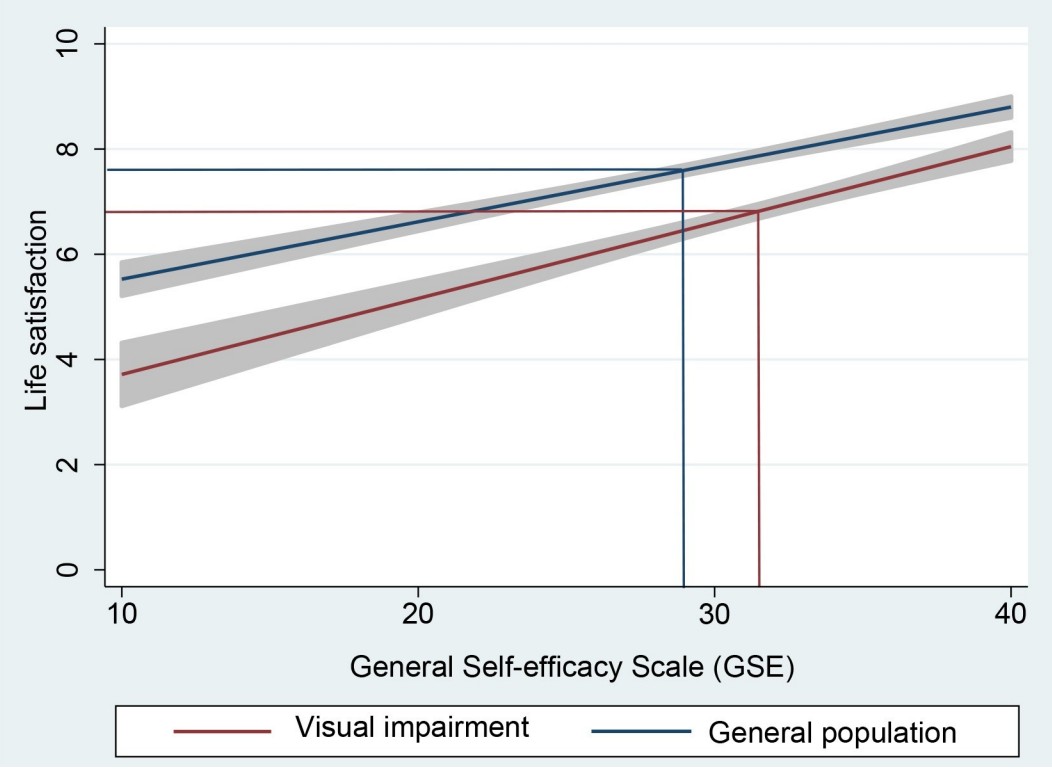

**Fig 2. Associations between general self-efficacy and life satisfaction in people with visual impairment and in the general population, presented as unadjusted linear predictions.** Gray area indicates 95% confidence intervals. The vertical lines indicate mean scores for self-efficacy, whereas the horizontal lines indicate mean scores for life satisfaction.

careers, social life and mobility, as well as their poorer access to information and non-verbal communication [26]. Even subtle aspects, that are often taken for granted by most sighted people, can present major challenges for individuals with impaired vision.

Based on our findings, it could be that people with visual impairment do not compare much with sighted people in terms of their ability to succeed. Rather, they have experienced mastery in coping with life based on their own preconditions. In line with the theories of Bandura [7, 8], we can postulate that self-efficacy has developed through successful experience in mastering challenges that sighted people do not typically have. This probably requires being part of a culture that recognizes such mastery. It probably also helps to have peers or be part of a peer community that can foster recognition. The finding that self-efficacy was not related to the degree of vision loss or the age at onset, suggests that self-efficacy may develop at any age and with many types of vision loss.

On the other hand, the finding that a lower degree of self-efficacy in those having other impairments in addition to vision loss indicates that additional disorders or disabilities can reduce the belief in one's ability to influence life—or the other way around, that lower self-efficacy may increase vulnerability to impaired health status [27]. Similar interpretations are relevant to the relationship between certain life stressors and self-efficacy. People with lower psychosocial resources like self-efficacy may be more vulnerable to exposure to violence or abuse—or, as shown in some studies, adverse life events, such as violence and abuse, can negatively impact self-efficacy [28, 29]. Childhood abuse, for instance, has been shown to negatively influence one's cognitions about the self [30].

## Life satisfaction

The linear relationship between self-efficacy and life satisfaction in the visual impairment population was similar to that in the general population as well as findings reported in previous studies [18], illustrating the significance of self-efficacy as an important element for a good life. This information has important implications for professionals and others who face people with vision loss. Overcoming obstacles and challenging situations may be a key factor for increasing self-efficacy. Therefore, rehabilitation programs should be individualized and facilitated with training tasks and sub-goals that are achievable. Feedbacks should be positive and help to increase motivation and experience of mastery.

The findings illustrate the need to warn against the ecological fallacy that may arise if one assumes that variables at the group level apply to associations between these variables at the level of the individual [31, 32]. Although self-efficacy was higher in people with visual impairment, life satisfaction was not. Life satisfaction was in fact significantly lower both when compared with a general population study that included an identical instrument of measurement [33], and when compared with the present general population sample in which we used an instrument with an almost identical wording.

In the extension of Bandura's theories [7, 8], both self-efficacy and life satisfaction may arise through the dynamic interaction between the person's characteristics, his or her behavior, and the environment in which the person lives and acts. Lower levels of life satisfaction among people with visual impairment may be due to restricted access to information, reduced mobility, lower education, more loneliness [2], and more adversities such as experiences with bullying [19], abuse [20], or mental disorders [3, 15]. In addition to improve the coping abilities of people with visual impairment, interventions should also target social or structural barriers in society, such as providing universal design and access to equal opportunities and physical environments, to increase the life satisfaction in this population.

## Strengths and limitations

Strengths of this study were a large probability sample of people with visual impairment of all ages, the use of telephone interviews to include individuals who would not otherwise respond, a validated instrument to assess general self-efficacy, and the inclusion of a comparable reference group of people from the general population. By oversampling younger adults, we were able to obtain robust estimates of self-efficacy and life satisfaction in a wide range of age groups.

Several limitations should be noted. Data were obtained by telephone interviews in the visual impairment population and by postal survey in the general population, a possibility of bias without a known direction. Second, we had limited information about the non-responders and do not know how non-responding might have influenced our results. Third, the inclusion of different measures of life satisfaction could have hampered the comparability of results between the two populations. However, the observed life satisfaction scores for the general population corresponds perfectly to that obtained in a Norwegian survey including an identical quality of life instrument as that used in our study of people with visual impairment [33]. We therefore believe that our findings for life satisfaction are reliable. Fourth, the cross-sectional study design limited the possibility to address relationships of cause and effect. Lastly, inclusion of participants from a membership organization for people who are blind or visually impaired questions whether the study sample was representative of the total population of people with visual impairment. Compared to 2015 census data from Statistics Norway [34], our study sample did not differ in terms of gender, employment, and place of residence, but had a higher level of education.

## Conclusions

Compared to the general population, people with visual impairment have a stronger belief in their ability to cope with challenges, which is referred to as self-efficacy. Within the population of people with visual impairment as well as in the general population, there was a linear relationship between self-efficacy and life satisfaction. Still, people with visual impairments had lower levels of satisfaction with their lives.

The higher self-efficacy among people with visual impairment may arise from extensive mastery experience in how to handle life as a person with reduced vision and constitutes a positive driving force in achieving the best possible life. Thus, provided that much in society is facilitated, the cultural minority of visually impaired should be encouraged to maintain an individual and collective attitude to practical problems as something that can be solved. Furthermore, they should be recognized for their ability to do so. Building self-efficacy in people with visual impairment can be essential to achieve the goal of a better life.

## Acknowledgments

Gratitude goes to the project partners in the European Network for Psychosocial Crisis Management—Assisting Disabled in Case of Disaster (EUNAD) and to the Norwegian Association of the Blind and Partially Sighted for making this study possible. We are also grateful to the collaborative project group of professionals and user representatives for their valuable feedback and discussions.

## Author Contributions

**Conceptualization:** Trond Heir.

**Data curation:** Audun Brunes.

**Formal analysis:** Audun Brunes, Trond Heir.

**Funding acquisition:** Audun Brunes, Trond Heir.

**Investigation:** Marianne B. Hansen, Trond Heir.

**Methodology:** Marianne B. Hansen, Trond Heir.

**Project administration:** Marianne B. Hansen, Trond Heir.

**Software:** Audun Brunes.

**Supervision:** Trond Heir.

**Validation:** Audun Brunes, Trond Heir.

**Visualization:** Audun Brunes, Trond Heir.

**Writing – original draft:** Audun Brunes, Marianne B. Hansen, Trond Heir.

**Writing – review & editing:** Audun Brunes, Marianne B. Hansen, Trond Heir.

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
