## [Decision Letter · Decision Letter 0]

17 May 2021

PONE-D-21-01130

General self-efficacy in individuals with visual impairment compared with the general population

PLOS ONE

Dear Dr. Brunes,

Thank you for submitting your manuscript to PLOS ONE. After careful consideration, we feel that it has merit but does not fully meet PLOS ONE’s publication criteria as it currently stands. Therefore, we invite you to submit a revised version of the manuscript that addresses the points raised during the review process. I have had an expert review it, and I have also read your paper and agree with all of the points and suggestions raised by the review. These are extensive, yet relatively minor, so please do address every point in a revised manuscript.

We look forward to receiving your revised manuscript.

Kind regards,

Michael J Proulx, Ph.D.

Academic Editor

PLOS ONE

Journal Requirements:

4.Thank you for stating the following in the Competing Interests section:

"I have read the journal's policy and the authors of this manuscript have the following competing interests: Research grants from the European Commission and research grants from the Norwegian Association of the Blind and Partially Sighted."

Reviewers' comments:

Reviewer's Responses to Questions

**Comments to the Author**

1. Is the manuscript technically sound, and do the data support the conclusions?

Reviewer #1: Yes

2. Has the statistical analysis been performed appropriately and rigorously? 

Reviewer #1: Yes

3. Have the authors made all data underlying the findings in their manuscript fully available?

Reviewer #1: Yes

4. Is the manuscript presented in an intelligible fashion and written in standard English?

Reviewer #1: Yes

5. Review Comments to the Author

Reviewer #1: This is an interesting paper that is looking at general self efficacy for people with visual impairment. The data were collected in 2017, and as should be noted as such. However, in saying this the sample size of the non control groups is worth noting at 736 individuals with a "self declared" notification with vision impairment as well as a large control 1792. Instruments were amended to account for degree of vision loss and age, as well as adding a QoL question.

Comorbidity

It is not clear in the paper the numbers that had an additional disability over and above VI (as well as the general population) You do state in the paper that comorbidity does have an effect (line 257 and line 279), and so is there any relationship or not with a particular kind of comorbidity? ( if the numbers allow for analysis).

Whilst it is acknowledged that people with no or low vision may have greater difficulties - it is not always the case - (line 262 ) perhaps insert the word "may" before the word face. Also I am not so sure that "Even subtle aspects, that are often taken for granted by most sighted (line 266) people, can present major challenges for individuals with impaired vision" adds anything to the discussion except for highlighting a more deficit account of vision impairment.

272 - normally have - replace with typically have

296-305 is really interesting paragraph and I think more could be discussed about the non relationship between self efficacy and life satisfaction, and what this means to people with VI.

338 please do not use phrases like "the blind"

Line 344...A lower life satisfaction...... this is too much of a leap - you may be right but perhaps could be removed from the conclusion and put into the discussion after all this is not a finding of the paper.

I am a little confused and perhaps could be explained more in the discussion around mastery. It seems from the data that mastery leading to greater self efficacy is found even within the earliest age group If mastery can be found in the younger group then perhaps what are the implications for practitioners? And should the scale be part of the took kit to determine if self efficacy has been "mastered". Does this finding soften your argument of mastery?

Note: visually impaired people (puts the disability first) people with vision impairment (puts people first) I prefer the later rather than the former please consider rewriting how you describe people with VI.

6. PLOS authors have the option to publish the peer review history of their article (what does this mean?). If published, this will include your full peer review and any attached files.

Reviewer #1: No

---

## [Author Response · Author response to Decision Letter 0]

24 May 2021

Response to Academic Editor

Response: We have checked the style requirements, especially those for file naming, and made the necessary changes.

Response: We have read through the reference list, and found some minor errors in the styling of citations. Apart from that, the reference list seemed complete and correct, and we did not identify any retracted papers.

3. We note that you have indicated that data from this study are available upon request. PLOS only allows data to be available upon request if there are legal or ethical restrictions on sharing data publicly. In your revised cover letter, please address the following prompts: a) If there are ethical or legal restrictions on sharing a de-identified data set, please explain them in detail (e.g., data contain potentially sensitive information, data are owned by a third-party organization, etc.) and who has imposed them (e.g., an ethics committee). Please also provide contact information for a data access committee, ethics committee, or other institutional body to which data requests may be sent.

Response: We thank the Editor for this comment. The present study contains sensitive information (e.g., health data) on a small group in society, especially those with blindness and those having congenital vision loss. Public availability may result in the possibility of indirect identification, and thus would compromise privacy of the participants. The right to privacy is sequred through for example Norway’s Personal Data Act. We have therefore made major revisions to the Data Availability Statement to clarify these issues:

Data are from the research project European Network for Psychosocial Crisis Management – Assisting Disabled in Case of Disaster (EUNAD). According to the approval from the Norwegian Regional Ethical Committee, data is to be stored properly and in line with the Norwegian privacy protection laws. Data contains sensitive information from a small group. Public availability may result in the possibility of indirect identification, and thus would compromise privacy of the participants. However, the data is freely available to interested researchers upon request, pending ethical approval from our ethical committee: post@helseforskning.etikk.no. The project leader, Prof. Trond Heir (trond.heir@medisin.uio.no), may also be contacted with requests for the data underlying our findings.

4.Thank you for stating the following in the Competing Interests section: "I have read the journal's policy and the authors of this manuscript have the following competing interests: Research grants from the European Commission and research grants from the Norwegian Association of the Blind and Partially Sighted." Please include your updated Competing Interests statement in your cover letter; we will change the online submission form on your behalf. Please confirm that this does not alter your adherence to all PLOS ONE policies on sharing data and materials, by including the following statement: "This does not alter our adherence to PLOS ONE policies on sharing data and materials.” (as detailed online in our guide for authors http://journals.plos.org/plosone/s/competing-interests). If there are restrictions on sharing of data and/or materials, please state these. Please note that we cannot proceed with consideration of your article until this information has been declared.

Response: We confirm our competing interests and declare that this does not alter our adherence to PLOS ONE policies on sharing data and materials. The updated competing interests statement reads as follows:

I have read the journal's policy and the authors of this manuscript have the following competing interests: Research grants from the European Commission and research grants from the Norwegian Association of the Blind and Partially Sighted. This does not alter our adherence to PLOS ONE policies on sharing data and materials.

Response to Reviewer

1. It is not clear in the paper the numbers that had an additional disability over and above VI (as well as the general population) You do state in the paper that comorbidity does have an effect (line 257 and line 279), and so is there any relationship or not with a particular kind of comorbidity? ( if the numbers allow for analysis).

Response: In the visual impairment population, the participants were first asked to indicate whether they had an additional impairment and then they were asked to describe in an open-ended response to what type of impairment(s) they had. Of the 258 individuals (35%) having other impairments in addition to the vision loss, about 75 percent of them reported one impairment and the remaining 25 percent reported two or more impairments. Most of the impairments listed by the participants were related to movement challenges and hearing loss.

For the visual impairment population, we obtained information on functional impairments, and did not address specific types of medical diagnoses, which was the case for the general population. Because of this, comparisons between the two populations in health status becomes inappropriate.

To specify the exact percentage of people from the visual impairment population having other impairments in addition to the vision loss, the following information was added to the second paragraph of the ‘Results’ section, line 201, page 10:

Roughly one in three (35%) participants had other impairments in addition to the vision loss. 

2. Whilst it is acknowledged that people with no or low vision may have greater difficulties - it is not always the case - (line 262 ) perhaps insert the word "may" before the word face. Also I am not so sure that "Even subtle aspects, that are often taken for granted by most sighted (line 266) people, can present major challenges for individuals with impaired vision" adds anything to the discussion except for highlighting a more deficit account of vision impairment.

Response: We agree that this is about the probability of facing great barriers to functioning and social participation, and that there are many people with visual impairment who are fully able to cope and self-manage in everyday life. We therefore followed the Reviewer’s suggestion, and added the term ‘may’ to this sentence (line 255-256, page 14-15).

3. 272 - normally have - replace with typically have

Response: We have replaced the term ‘normally’ with the term ‘typically’.

4. 296-305 is really interesting paragraph and I think more could be discussed about the non relationship between self efficacy and life satisfaction, and what this means to people with VI.

Response: We are glad to hear that the Reviewer found our discussion interesting. The main focus of the discussion has been to explain why people with visual impairment have a higher self-efficacy relative to the general population, but still score lower on life satisfaction. It could mean that building self-efficacy are important but insufficient, and that there is a need of interventions that also targets social and structural barriers in the society to improve these people’s life satisfaction. To highlight this point, we have therefore made some revisions in the ‘Discussion’ section, page 16:

In the extension of Bandura's theories [7,8], both self-efficacy and life satisfaction may arise through the dynamic interaction between the person's characteristics, his or her behavior, and the environment in which the person lives and acts. Lower levels of life satisfaction among people with visual impairment may be due to restricted access to information, reduced mobility, lower education, more loneliness [2], and more adversities such as experiences with bullying [19], abuse [20], or mental disorders [3,15]. In addition to improve the coping abilities of people with visual impairment, interventions should also target social or structural barriers in society, such as providing universal design and access to equal opportunities and physical environments, to increase the life satisfaction in this population.

building self-efficacy among people with visual impairment is important but insufficient, and that interventions targeting social and structural factors in society, such as universal design and access to physical environments and equal opportunities, are also needed to improve these people’s life satisfaction.

5. 338 please do not use phrases like "the blind"

Response: We have made extensive revisions and are now using a person-first language when describing the visual impairment population throughout the entire paper.

6. Line 344...A lower life satisfaction...... this is too much of a leap - you may be right but perhaps could be removed from the conclusion and put into the discussion after all this is not a finding of the paper.

Response: We agree. The sentence (line 344, page 18) has been removed from the conclusions.

7. I am a little confused and perhaps could be explained more in the discussion around mastery. It seems from the data that mastery leading to greater self efficacy is found even within the earliest age group If mastery can be found in the younger group then perhaps what are the implications for practitioners? And should the scale be part of the took kit to determine if self efficacy has been "mastered". Does this finding soften your argument of mastery?

Response: We hypothesize that mastery experiences may explain the high levels of self-efficacy among people with visual impairment. Because self-efficacy is something that can be built, and our findings of a linear relationship between self-efficacy and life satisfaction among people with visual impairment, this may have implications for practitioners. We have therefore included the following information in the ‘Discussion’ section, on page 16.

The linear relationship between self-efficacy and life satisfaction in the visual impairment population was similar to that in the general population as well as findings reported in previous studies [18], illustrating the significance of self-efficacy as an important element for a good life. This information has important implications for professionals and others who face people with vision loss. Overcoming obstacles and challenging situations may be a key factor for increasing self-efficacy. Therefore, rehabilitation programs should be individualized and facilitated with training tasks and sub-goals that are achievable. Feedbacks should be positive and help to increase motivation and experience of mastery.

8. Note: visually impaired people (puts the disability first) people with vision impairment (puts people first) I prefer the later rather than the former please consider rewriting how you describe people with VI.

Response: We thank the reviewer for highlighting this, and hope that the reviewer finds our efforts in using a person-first language satisfactory.

---

## [Decision Letter · Decision Letter 1]

21 Jun 2021

General self-efficacy in individuals with visual impairment compared with the general population

PONE-D-21-01130R1

Dear Dr. Brunes,

We’re pleased to inform you that your manuscript has been judged scientifically suitable for publication and will be formally accepted for publication once it meets all outstanding technical requirements.

Kind regards,

Michael J Proulx, Ph.D.

Academic Editor

PLOS ONE

Additional Editor Comments (optional):

Reviewers' comments:

Reviewer's Responses to Questions

**Comments to the Author**

1. If the authors have adequately addressed your comments raised in a previous round of review and you feel that this manuscript is now acceptable for publication, you may indicate that here to bypass the “Comments to the Author” section, enter your conflict of interest statement in the “Confidential to Editor” section, and submit your "Accept" recommendation.

Reviewer #1: All comments have been addressed

2. Is the manuscript technically sound, and do the data support the conclusions?

Reviewer #1: Yes

3. Has the statistical analysis been performed appropriately and rigorously? 

Reviewer #1: Yes

4. Have the authors made all data underlying the findings in their manuscript fully available?

Reviewer #1: (No Response)

5. Is the manuscript presented in an intelligible fashion and written in standard English?

Reviewer #1: Yes

6. Review Comments to the Author

Reviewer #1: Having gone through the paper now I am happy that you have address all of my concerns that were initially outlined in the first review. There are some particular interesting findings within this paper and that I am also please you have focused on a less deficit account.

7. PLOS authors have the option to publish the peer review history of their article (what does this mean?). If published, this will include your full peer review and any attached files.

Reviewer #1: **Yes: **John Ravenscroft

---

## [Editor Report · Acceptance letter]

24 Jun 2021

PONE-D-21-01130R1 

General self-efficacy in individuals with visual impairment compared with the general population 

Dear Dr. Brunes:

I'm pleased to inform you that your manuscript has been deemed suitable for publication in PLOS ONE. Congratulations! Your manuscript is now with our production department. 

Kind regards, 

on behalf of

Dr. Michael J Proulx 

Academic Editor

PLOS ONE